# Synthesis and Characterization of a New Norfloxacin/Resorcinol Cocrystal with Enhanced Solubility and Dissolution Profile

**DOI:** 10.3390/pharmaceutics14010049

**Published:** 2021-12-27

**Authors:** Hanan Fael, Rafael Barbas, Rafel Prohens, Clara Ràfols, Elisabet Fuguet

**Affiliations:** 1Departament d’Enginyeria Química i Química Analítica, Universitat de Barcelona, Martí i Franquès 1–11, 08028 Barcelona, Spain; hananfael@hotmail.com (H.F.); crafols@ub.edu (C.R.); 2Unitat de Polimorfisme i Calorimetria, Centres Científics i Tecnològics, Universitat de Barcelona, Baldiri Reixac 10, 08028 Barcelona, Spain; rafa@ccit.ub.edu

**Keywords:** Norfloxacin, resorcinol, cocrystal, cocrystallization, dissolution rate, solubility

## Abstract

A new cocrystal of Norfloxacin, a poorly soluble fluoroquinolone antibiotic, has been synthetized by a solvent-mediated transformation experiment in toluene, using resorcinol as a coformer. The new cocrystal exists in both anhydrous and monohydrate forms with the same (1:1) Norfloxacin/resorcinol stoichiometry. The solubility of Norfloxacin and the hydrated cocrystal were determined by the shake-flask method. While Norfloxacin has a solubility of 0.32 ± 0.02 mg/mL, the cocrystal has a solubility of 2.64 ± 0.39 mg/mL, approximately 10-fold higher. The dissolution rate was tested at four biorelevant pH levels of the gastrointestinal tract: 2.0, 4.0, 5.5, and 7.4. In a first set of comparative tests, the dissolution rate of Norfloxacin and the cocrystal was determined separately at each pH value. Both solid forms showed the highest dissolution rate at pH 2.0, where Norfloxacin is totally protonated. Then, the dissolution rate decreases as pH increases. In a second set of experiments, the dissolution of the cocrystal was evaluated by a unique dissolution test, in which the pH dynamically changed from 2.0 to 7.4, stepping 30 min at each of the four biorelevant pH values. Results were quite different in this case, since dissolution at pH 2 affects the behavior of Norfloxacin at the rest of the pH values.

## 1. Introduction

The systemic effect of a drug starts with drug absorption, which in turn consists of a succession of rate processes: the release of the drug from the drug product, dissolution of the drug in an aqueous environment, and absorption across cell membranes into the systemic circulation. For drugs that have very poor aqueous solubility, the rate at which the drug dissolves (dissolution rate) is often the slowest step and therefore exerts a rate-limiting effect on drug bioavailability [1]. Nowadays, the low solubility is a major drawback related to drugs coming from synthesis, and this has a direct impact on their bioavailability [2]. To overcome this problem, diverse strategies have been developed. These include changing the crystal form (polymorphs, hydrates, etc.), formation of an amorphous phase, salt formation, formulation of solid dispersions with hydrophilic polymers [3], and inclusion of a complex formation with cyclodextrins [4]. Cocrystallization is an evolving technique that can dramatically modify the physical properties of drugs, such as stability, melting point, solubility, and dissolution rate, without altering the pharmacological effect of the drug candidate [5,6,7,8]. Therefore, increasing attention has been paid to cocrystallization as a tool to improve the pharmaceutical properties of drugs in the pharmaceutical industry and academic fields. A cocrystal can be defined as “a stoichiometric multi-component crystal form connected by non-covalent interactions where all the components present are solid under ambient conditions” [6,9,10]. Thus, a pharmaceutical cocrystal is composed of an active pharmaceutical ingredient (API) and a suitable organic and safe compound called the coformer [11]. There are several techniques to prepare cocrystals, such as solution crystallization (solvent evaporation/cooling) [12,13], mechanical grinding—in the absence of a solvent or in the presence of a minor amount of liquid [14]—and melt crystallization [15]. Norfloxacin (Figure 1) is a poorly soluble BCS class IV fluoroquinolone antibiotic [16]. It is a broad-spectrum antibiotic, effective against both Gram-positive and Gram-negative aerobic bacteria, and thus it is mainly used in the treatment of bacterial infections of the lower respiratory and urinary tracts [17]. Norfloxacin has relatively low oral bioavailability (~40%), limited by its poorly aqueous solubility [18,19]. The solid-state landscape of Norfloxacin was reviewed by some of us in 2010 [20], with only limited studies being reported on the cocrystallization of the drug: the sacharinate–sacharine dihydrate cocrystal [21], the isonicotinamide–chloroform cocrystal/solvate with a solubility increase of three-fold compared to the plain drug [20], and, most recently, the nicotinic acid cocrystal with solubility two-times greater than Norfloxacin [22].

With the aim of improving the solubility and dissolution rate of Norfloxacin in the form of a pharmaceutical cocrystal, and considering these antecedents, resorcinol (Figure 1) was selected as the coformer in this work. It is a versatile compound with two strong hydrogen bond donor groups, with the potential to interact with both the carbonyl and amine hydrogen bond acceptor groups of Norfloxacin. Moreover, it shows important antioxidant properties, and its FEMA GRAS status [23], granted by the European Food Safety Authority [24], enables resorcinol to serve as an authorized flavoring substance without “safety concern at estimated levels of intake as flavouring substances”. In fact, some of us have studied this coformer in other crystal engineering studies [25,26], with the results demonstrating its remarkable ability to improve the stability of vitamin D_3_ via cocrystallization [27]. Resorcinol has also been previously utilized to prepare cocrystals with agomelatine [28], curcumin [29], and Ciprofloxacin, a fluoroquinolone antibiotic similar to Norfloxacin [30].

## 2. Materials and Methods

### 2.1. Materials

Norfloxacin (>98%), resorcinol (>99%), and toluene (>99.5%) were purchased from Sigma Aldrich (Darmastadt, Germany). Norfloxacin was a mixture of polymorph A, polymorph C, and the sesquihydrate Form I (Ref Code PUZGAT). See Appendix A for further details [31].

The following chemicals were used to prepare buffer solutions: ethylenediamine (>98.8%) from J. T. Baker (Radnor, PA, USA), and trifluoroacetic acid (>99%) and acetic acid (>99%) from Merck (Darmastadt, Germany). To reach the desired pH value, 1 M sodium hydroxide Titrisol^®^ or 0.5 M hydrochloric acid Titrisol^®^, also from Merck, were added.

For the dissolution rate determination, the following reagents were used: dimethyl sulfoxide (>99.9%, DMSO), potassium dihydrogen phosphate (>99.5%), 0.5 M potassium hydroxide Titrisol^®^, and 0.5 M hydrochloric acid Titrisol^®^, all from Merck. Potassium chloride (>99%) was from Sigma.

For HPLC quantification, acetonitrile (HPLC gradient-grade) from Fisher Scientific (Hampton, NH, USA) and a mixture of sodium dihydrogen phosphate monohydrate (>99%) and disodium monohydrogen phosphate (>99%) from Merck were used.

Water was purified by a Milli-Q plus system from Millipore (Bedford, MA, USA), with a resistivity of 18.2 MΩ cm.

### 2.2. Synthesis of the Cocrystal

Synthesis of the anhydrous Norfloxacin–resorcinol cocrystal was conducted by solvent-mediated transformation. In particular, Norfloxacin (95.8 mg, 0.300 mmol) and resorcinol (39.6 mg, 0.360 mmol) were stirred in toluene (2.0 mL) at room temperature (25 °C) overnight. The resulting suspension was filtered and dried under vacuum. Yield = 70%. The Norfloxacin–resorcinol cocrystal monohydrate was isolated during the solubility/dissolution rate determination experiments.

### 2.3. Differential Scanning Calorimetry (DSC)

Differential scanning calorimetry measurements were carried out by means of a Mettler Toledo (Columbus, OH, US) DSC-822e calorimeter. Experimental conditions: aluminum crucibles of 40 μL, dry nitrogen with 50 mL/min flow rate, heating from 30 °C to 300 °C at a rate of 10 °C/min. The calorimeter was calibrated with indium of 99.99% purity (m.p.: 156.6 °C; ΔH: 28.32 J/g).

### 2.4. Thermogravimetric Analysis (TGA)

Thermogravimetric measurements were performed on a Mettler Toledo TGA-851e thermobalance (Columbus, OH, US). Experimental conditions: alumina crucibles of 70 μL, dry nitrogen with 50 mL/min flow rate, heating from 30 °C to 300 °C at a rate of 10 °C/min. A blank curve was previously performed using the same methodology and it was subtracted.

### 2.5. Nuclear Magnetic Resonance (NMR)

Proton nuclear magnetic resonance (^1^H-NMR) spectra were measured on a Varian Mercury 400 (400 MHz) (Santa Clara, CA, US) spectrometer. Chemical shifts for proton are reported in parts per million (ppm) downfield from tetramethylsilane (TMS) and referenced to the residual proton signal in the NMR solvent (dmso-*d*_6_: δ 2.50). Experimental conditions, delay: 1 s; pulse: 45°; scans: 16 or 32.

### 2.6. X-ray Crystallographic Analysis

X-ray powder diffraction (XRPD) patterns of new cocrystals of Norfloxacin were measured on a PANalytical (Malvern, UK) X’Pert PRO MPD (transmission configuration with Cu Kα1+2 radiation, λ = 1.5406 Å) with a focalizing elliptic mirror and a PIXcel detector, and a maximum active detector length of 3.347°. The transmission geometry configuration included a convergent beam with a focalizing mirror, a flat sample sandwiched between low absorbing films measuring from 1 to 40° in 2θ, a step size of 0.026°, and a measuring time of 30 min at 298 K.

### 2.7. Solubility

Solubility measurements were performed following the consensus recommendations for the shake-flask method described elsewhere [32]. A universal buffer composed of 75 mM trifluoroacetic acid, 25 mM acetic acid, and 25 mM ethylenediamine, adjusted with 1M sodium hydroxide up to pH 7.5, was used (ionic strength, I = 0.1 M). Samples were prepared by adding a weighted amount of solid (enough to obtain saturated solutions) to 2 mL of the buffered solution. After this, solutions were shaken at a controlled temperature (25.0 ± 0.2 °C) for 24 h, but, after 4 h of agitation, the pH was checked and, if necessary, readjusted to the initial value with small amounts of 1 M hydrochloric acid or 1 M sodium hydroxide solution. After the 24 h stirring, samples were left to stand for equilibration for 24 more hours [32,33,34]. The pH was measured and phases separated by centrifugation in a Rotanta 460RS centrifuge from Hettich Lab Technology (Tuttlingen, Germany), at a controlled temperature (25 °C) and 3500 rpm for 30 min.

The remaining solid was dried under vacuum for 30 min and stored at 4 °C until XRPD analysis. Solubility was obtained by quantifying the concentration of Norfloxacin and resorcinol in the supernatant by liquid chromatography. A Shimadzu (Kyoto, Japan) liquid chromatograph, equipped with two LC-10AD pumps and a SPD-10AV detector, was used. Temperature was controlled at 25.0 ± 0.1 °C with a CTO-10AS column oven. The column was a Symmetry C18, with 5 μm particle size and dimensions of 4.6 × 150 mm from Waters (Milford, MA, USA). Mobile phase was composed of 20 mM phosphate buffer at pH 7 (A) and acetonitrile (B). Elution was in gradient mode (time (min), %B: 0,3; 1,3; 5,25; 10,25; 12,3; 15,3). Flow rate was 1 mL/min, the injection volume was 10 μL, and the detection wavelengths were set at 254 nm for resorcinol and 267 nm for Norfloxacin. To quantify, six standard solutions were prepared in the linear range (5–60 mgL^−1^ for both compounds). Standards were dissolved in an acetonitrile/water mixture at a proportion of 25/75 (*v*/*v*). In a first attempt, saturated supernatants were injected directly, and, when necessary, they were appropriately diluted in water to ensure that they fit within the range of calibration. The calibration curve for resorcinol was A = 37217C-3717 (R^2^ = 0.9988), and the calibration curve for Norfloxacin was A = 33554C-29994 (R^2^ = 0.9991). None of the samples had concentrations below the most diluted standards (5 mg mL^−1^).

### 2.8. Intrinsic Dissolution Rate (IDR)

Molar extinction coefficients of Norfloxacin and resorcinol were determined by UV-metric titration using a GlpKaTM titrator (Sirius Analytical Instruments, Forest Row, UK). Briefly, a 10 mM stock solution of each compound was prepared in DMSO. Then, 50 μL of sample stock solution and 0.25 mL of 15 mM potassium phosphate buffer were added to 10 mL of 0.15 M KCl solution. The pH was adjusted to 2 with 0.5 M HCl before starting the titration, and then the titration was done using 0.5 M KOH up to pH 12. The UV absorption spectra (between 250 and 400 nm) of the solution were recorded at each titrant addition by a fiber optic dip-probe. The collected data were refined through the Sirius T3 v. 2.0.0 software, and the p*K_a_* values and Molar Extinction Coefficients obtained by Target Factor Analysis (Appendix A). The plot shows that the molar absorption of resorcinol was much lower than that of Norfloxacin in the whole studied wavelength range. In any case, quantification of Norfloxacin concentration was done from 300 to 400 nm, a range in which resorcinol does not show absorbance.

Tablets of 3 mm diameter containing 7–9 mg of pure drug or Norfloxacin–resorcinol cocrystal were prepared using a manual hydraulic press (Applied Measurements Ltd., Reading, UK), applying a constant pressure of 0.1 ton for 2 min. Dissolution tests were performed with a small-scale dissolution assay installed in the GlpKaTM titrator. The tablet was placed in a holder, and only one side of the tablet was exposed to the dissolution medium, with a total exposed surface area of 0.07 cm^2^. Then, 1.5 mL of 0.125 M acetate–phosphate buffer at pH 1.6 was introduced into the vial without wetting the tablet surface. The instrument then automatically added a suitable volume of 0.15 M KCl solution and adjusted the pH to the required value. The final volume was 15 mL. UV–visible absorption spectra were recorded every 30 s with a fiber-optic probe immersed in the solution.

To perform the 4-sector dissolution rate determinations, the same procedure was followed, but, after adjusting the initial pH to 2.0 and monitoring dissolution for 30 min at this pH, the pH was automatically raised to the next value by addition of KOH. Then, dissolution was monitored again for 30 more minutes. This stepping procedure was repeated until dissolution in the four sectors was covered.

The concentration of Norfloxacin in the solution at each time point was determined from the spectroscopic data by applying the Beer–Lambert law, using the previously determined molar extinction coefficients. Spectral regions where the signal was saturated (A > 1.5) were discarded. Dissolution rate was obtained through the fitting of the first-order Noyes–Whitney exponential equation to the data:(1)$${\left[X\right]}_{t}=S\left(1-e^{-k_{d}\left(t-t_{0}\right)}\right)$$

In this equation, [*X*]*_t_* is the weight (g) of compound in solution at a given time (min); *S* is the extrapolated solubility (g), and *k_d_* is the dissolution rate constant (min^−1^). *t*_0_ (min) is a term allowing for a temporal offset. Dissolution rate is calculated through a refinement process in which *S*, *k_d_* and *t*_0_ optimal values are obtained by minimizing the root mean square deviation between the modeled concentrations and the measured ones. Then, the dissolution rate (g min^−1^) is obtained by the product *k_d_S* [35].

### 2.9. pK_a_ Determination

First, 5–6 mg of compound were dissolved in 15 mL of a 0.15 M KCl aqueous solution. The sample was pre-alkalinized to pH 12 with 0.5 M standardized KOH. Then, it was titrated with standardized 0.5 M HCl. All titrations were carried out at 0.15 M ionic strength and a temperature of 25 °C, under a nitrogen atmosphere. The p*K_a_* values were calculated through the Sirius T3 v. 2.0.0 software. The Debye–Hückel equation was used to obtain the thermodynamic p*K_a_* values.

## 3. Results and Discussion

### 3.1. Characterization of the Cocrystals

The Norfloxacin–resorcinol cocrystal was obtained in two different crystal forms, one anhydrous and one monohydrate with the same (1:1) API/coformer stoichiometry, based on NMR and TGA measurements and further characterized by means of XRPD and DSC. The XRPD diffractogram of the anhydrous Norfloxacin–resorcinol cocrystal was indexed with the following proposed triclinic cell: a = 18.427(5) Å, b = 14.593(3) Å, c = 19.613(6) Å, α = 44.30(2)°, β= 139.73(2)°, γ = 133.25(1)° and V = 2265(1) Å^3^ by means of Dicvol04 [36] (Figures of Merit: M = 11, F = 31) with a number of impurities equal to zero. The cell volume is compatible with four molecules of Norfloxacin and four molecules of resorcinol, Z = 4 (Figure 2).

The powder diffractogram of the Norfloxacin–resorcinol cocrystal monohydrate was indexed with the following proposed monoclinic cell: a = 18.336(2) Å, b = 7.8407(7) Å, c = 15.577(4) Å, β = 109.580(9)° and V = 2110.0(6) Å^3^ by means of Dicvol04 (Figures of Merit: M = 54, F = 165) with a number of impurities equal to zero. The *P*2*_1_*/*n* space group was determined based on the assessment of systematic absences and the cell volume is compatible with four molecules of Norfloxacin, four molecules of resorcinol and four molecules of water, Z = 4 (Figure 3).

On the other hand, the DSC analysis of the bulk anhydrous powder of the Norfloxacin–resorcinol cocrystal showed an endothermic phenomenon at 198 °C with an associated heat of 114.4 J/g (Appendix A). The TGA analysis did not show a weight loss before melting (Appendix A). Its powder diffractogram was indexed and the cell volume is compatible with one molecule of Norfloxacin and one molecule of resorcinol in the asymmetric unit. On the other hand, the DSC analysis of the bulk monohydrate powder of the Norfloxacin–resorcinol cocrystal showed a first wide endothermic phenomenon starting at 62 °C with an associated heat of 124.3 J/g, a second endothermic phenomenon at 124 °C with an associated heat of 3.7 J/g, followed by an exothermic phenomenon with an associated heat of 36.5 J/g and, finally, a third endothermic phenomenon at 189 °C with an associated heat of 128.6 J/g, (Appendix A). Moreover, and with the aim of characterizing the melting point of the monohydrate cocrystal, a DSC experiment was run at a 10 °C/min heating rate without a hole in the crucible. The thermogram showed a first endothermic phenomenon at 147 °C with an associated heat of 66.0 J/g and a second endothermic phenomenon at 186 °C with and associated heat of 141.7 J/g (Appendix A). The TGA analysis showed a weight loss of 3.8% detected from 30 to 138 °C, which can be attributed to one molecule of water per molecule of Norfloxacin and resorcinol (theoretical weight loss of 3.9%) (Appendix A). Its powder diffractogram was indexed and the cell volume is compatible with one molecule of Norfloxacin, one molecule of resorcinol and one molecule of water in the asymmetric unit. XRPD diffractograms of this multicomponent system compared to pure Norfloxacin and resorcinol are shown in Figure 4, and characteristic 2theta peaks of each form can be found in Appendix A. Finally, the ^1^H-NMR spectra of the obtained cocrystals were measured and the resonance peaks integrated to determine the API/conformer stoichiometric ratio in each cocrystal. Upon quantitative analysis of the ^1^H-NMR spectra, both Norfloxacin–resorcinol cocrystals were found to have a 1:1 molar ratio (see ESI for further characterization details).

### 3.2. Solubility Study

The solubility of Norfloxacin and the solubility product of the anhydrous cocrystal were measured by the shake-flask method [32,33,34]. Norfloxacin is a diprotic zwitterionic compound with p*K_a_* values in water of 6.35 ± 0.04 and 8.80 ± 0.04 (25 °C and 0 ionic strength, determined by potentiometry) [37]. Therefore, it will be in its ionic form at low (cation) and high (anion) pH values. However, as the two p*K_a_* values are quite close, a pH region in which only the zwitterionic form is present in the solution does not exist. The maximum percentage of the zwitterionic form (around 90%) will be obtained at a pH in between the two p*K_a_*s, i.e., 7.5 (see Appendix A). This is also the pH at which Norfloxacin is expected to show the lowest solubility.

In the case of a zwitterionic compound such as Norfloxacin, a saturated solution can be defined by the following equilibria and the related constants:(2)$$H_{2}X^{+}↔{\mathrm{HX}}^{\pm }+H^{+}\;\;\;\;\;\;\;\;\;K_{a1}=\frac{\left[{\mathrm{HX}}^{\pm }\right]\left[H^{+}\right]}{\left[H_{2}X^{+}\right]}$$
(3)$${\mathrm{HX}}^{\pm }↔X^{-}+H^{+}\;\;\;\;\;\;\;\;\;K_{a2}=\frac{\left[X^{-}\right]\left[H^{+}\right]}{\left[{\mathrm{HX}}^{\pm }\right]}$$
(4)$${\mathrm{HX}}_{\left(s\right)}^{\pm }↔{\mathrm{HX}}^{\pm }\;\;\;\;\;\;\;\;\;S_{0}=\left[{\mathrm{HX}}^{\pm }\right]$$
where *S*_0_ is the solubility of the neutral species—in the present case, the zwitterionic one. The total concentration of Norfloxacin in the solution is the sum of the concentrations of all the species dissolved in the aqueous phase, given by the mass balance:(5)$$C_{\mathrm{Nor}}=\left[H_{2}X^{+}\right]+\left[{\;\mathrm{HX}}^{\pm }\right]+\left[X^{-}\right]$$

In a saturated solution, the total concentration is the solubility, *S*, so Equation (5) is converted to:(6)$$S_{\mathrm{Nor}}=\left[H_{2}X^{+}\right]+\left[{\;\mathrm{HX}}^{\pm }\right]+\left[X^{-}\right]$$

It is useful to convert the above equation into an expression containing only constants and [H^+^] as the only variable, by substituting the ionization and solubility Equations (2)–(4) into Equation (6). Then, Equation (7) is obtained:(7)$$\log S_{Nor}=\log S_{0}+\log\left(10^{pK_{1}-\mathrm{pH}}+10^{\mathrm{pH}-pK_{2}}+1\right)$$

The chromatographic quantification of the saturated solutions after the shake-flask process provides a solubility for Norfloxacin of 0.394 ± 0.018 mg/mL (*n* = 3, final pH 7.25 after the shake-flask process). From this solubility value and Equation (7), it is possible to calculate the intrinsic solubility for the zwitterionic form (*S*_0_), which is 0.319 ± 0.019 mg/mL. Note that, as the ionic strength in solubility determinations is 0.1M, in this latter calculation, p*K_a_* values have been converted to p*K_a_*′ values (acidity constant at the working ionic strength, where hydrogen ion is expressed as activity and the rest of the species as concentrations) [38]. This *S*_0_ value is in agreement with literature values. Avdeef reported the average of 18 *S*_0_ determinations from different sources, in all cases performed by the shake-flask method, with *S*_0_ values ranging from 0.29 to 0.61 mg/mL [39].

The solubility of the anhydrous Norfloxacin–resorcinol cocrystal was calculated in a similar way. Resorcinol is a very weak diprotic acid. The first ionization has a p*K_a_* of 9.47 ± 0.02 (25 °C and 0 ionic strength, determined by potentiometry), and the second p*K_a_* is even higher (11.60 ± 0.04) [37]. For the cocrystal, the solubility product can be defined as:(8)$${\mathrm{HX}}^{\pm }\cdot {\mathrm{HA}}_{\left(s\right)}\;↔{\;\mathrm{HX}}^{\pm }+\mathrm{HA}\;\;\;\;\;\;\;\;\;K_{ps}=\left[{\;\mathrm{HX}}^{\pm }\right]\left[\mathrm{HA}\right]$$

In this equilibrium, HA stands for resorcinol, which, at the working pH of the present study (7.5), can be treated as a monoprotic acid. Similar to the solubility equilibrium of a single compound, in a cocrystal, the solid cocrystal is in equilibrium with the two components in the solution. The acid-base equilibrium of resorcinol can be expressed as
(9)$$\mathrm{HA}\;↔{\;A}^{-}+H^{+}\;\;\;\;\;\;\;\;\;K_{a}=\frac{\left[A^{-}\right]\left[H^{+}\right]}{\left[\mathrm{HA}\right]}$$

Moreover, the total concentration of resorcinol in the solution is given by the mass balance through Equation (10):(10)$$C_{\mathrm{Res}}=\left[\;\mathrm{HA}\right]+\left[A^{-}\right]$$

Combination of Equations (9) and (10) allows the calculation of the species [HA] at any pH value, simply quantifying the total amount of resorcinol in the solution, *C*_Res_ (Equation (11)):(11)$$\left[\mathrm{HA}\right]=\frac{C_{\mathrm{Res}}}{1+10^{\mathrm{pH}-pK_{a}}}$$

Similarly, combination of Equations (2) and (3) into Equation (5) provides the following expression, which allows the calculation of the concentration of [HX^±^] at any pH value, simply quantifying the total concentration of Norfloxacin in the solution (*C*_Nor_):(12)$$\left[{\mathrm{HX}}^{\pm }\right]=\frac{C_{\mathrm{Nor}}}{10^{pK_{a1}-\mathrm{pH}}+10^{\mathrm{pH}-pK_{a2}}+1}$$

The substitution of Equations (11) and (12) into Equation (8) allows the calculation of the solubility product of the cocrystal, which has a value of 3.85 × 10^−5^ ± 1.20 × 10^−5^ (*n* = 4). Then, the solubility of the cocrystal is easily calculated:(13)$$S=\sqrt{K_{ps}}$$

The solubility of the Norfloxacin–resorcinol cocrystal is 2.64 ± 0.39 mg/mL, almost 10-fold higher than the solubility of the API alone.

XRPD analysis of the solid phase obtained after the shake-flask experiments stated that there was a transformation of both solid forms during the solubility tests. Commercial Norfloxacin (which was a mixture of polymorph A, polymorph C, and the sesquihydrate Form I (Ref Code PUZGAT)) was transformed during the process to a tetrahydrate (according to TGA analysis and XRPD comparison with known forms of Norfloxacin; see Appendix A), so, in fact, the solubility values provided in this section correspond to the tetrahydrate form. In the case of the cocrystal, the anhydrous form was converted into a new form, which was identified and characterized in Section 3.1 as a monohydrate. Therefore, it is not possible to obtain the solubility of the initial solid form in either of the two cases, since transformations occur during the course of the experiments in contact with aqueous solutions.

### 3.3. Intrinsic Dissolution Rate Study

The dissolution rates of Norfloxacin and the anhydrous Norfloxacin–resorcinol cocrystal were determined at four different pH values (2.0, 4.0, 5.5, and 7.4). These values were chosen because they can be encountered in different parts of the gastrointestinal tract (GIT), so they should be adequate to evaluate the behavior of the compounds after oral intake. Figure 5 shows the dissolution profiles obtained at each pH, and Table 1 shows the dissolution percentage reached after 30 min of dissolution, together with the dissolution rate values obtained through the fitting of Equation (1) to the experimental data. Table 1 also shows the solid form identity results corresponding to the characterization of, on one hand, the initial solid forms and, on the other hand, the solid obtained after the dissolution experiments.

Figure 5A shows the dissolution profiles at pH 2. The initial slope of both curves is practically coincident, so Norfloxacin dissolves at the same rate independently of the solid form in which it exists. However, after 5 min, the dissolution rate of the cocrystal tends to be slightly lower, as the average slope decreases. This difference can be related to the presence of resorcinol in the solution. However, it must be noted that the dissolution experiments of Norfloxacin at this pH are very reproducible, whereas the error bars indicate high variability in the dissolution of the cocrystal in the time range of 0–15 min (sometimes, it dissolves similarly to the API, and, sometimes, the process is slightly slower). The lower reproducibility is mainly attributed to the different physical properties of the two tablets, since the Norfloxacin and cocrystal powders are physically different. In any case, the percentage of dissolution is very high in both cases (100% for Norfloxacin alone, 94% for the cocrystal). This is expected since, at such low pH, the API is in its cationic form.

Particularly interesting are the dissolution profiles obtained at pH 4.0 (Figure 5B). According to the Norfloxacin p*K_a_* values, a similar profile as the one at pH 2.0 would be expected (see Appendix A). However, the slope decreases markedly from pH 2.0 to pH 4.0, especially in the API curve. Compared to pH 2, the percentage of dissolved Norfloxacin after 30 min of dissolution is similar in the cocrystal tablet, whereas it falls to 71% in the API tablet. This behavior has been reported for some ionizable APIs—Ciprofloxacin, among others [30]—and can be explained by the change in pH at the tablet–solution interface (pH_x=0_ effect) [40]. When a large amount of API is dissolved, a saturated API solution is created only at the interface with the tablet surface. As a result, the pH at the interface is shifted according to the acid-base nature of the API, because the high concentration of API hinders the buffer’s capacity to keep the pH constant.

To prove this, a set of vials with dissolution media at pH 4.0, 5.5, and 7.4 were prepared (identical amount as in the dissolution rate experiments). However, instead of adding a Norfloxacin tablet, the powder was directly weighted into the vials until saturated solutions of Norfloxacin were obtained. Then, the solutions were allowed to stand under agitation at 25 °C for 24 h. After this, the pH of the solutions was measured again. The new pH values were 5.8, 6.2, and 7.4, respectively, i.e., there were positive pH increments of 1.8, 0.7, and 0 units, respectively. This additional test simulates what is happening at the interface, revealing a pH higher than the one of the bulk solution, and causing a decrease in the dissolution process. This effect is more evident for the API dissolution than for the cocrystal because the amount of API released in the cocrystal is lower compared to the pure API.

The same phenomenon is observed in the profiles at pH 5.5. At this pH, there is part of Norfloxacin in the zwitterionic form already, so a decrease in dissolution rate is expected. Nevertheless, the pH_x=0_ effect can be also noticed, although at a lower degree, and the cocrystal dissolves faster than the API. Notice that, at pH 4 and pH 5.5, there is no overlapping in the error bars, so dissolution is clearly different for the two solid forms. In the cocrystal, dissolution of 48% of the API is observed after 30 min, whereas the percentage in solution achieved with API tablets is only 23%.

Finally, the profiles at pH 7.4 show the same behavior for the API and the cocrystal. Here, the amount of API in the solution decreases to 7%. In fact, at this pH, the solubility of Norfloxacin has the minimum value, which is, in some way, reflected in the dissolution curves. Moreover, this pH is not affected by the pH_x=0_ phenomenon since this is the pH provided by an unbuffered Norfloxacin saturated solution, so, in this particular case, the cocrystal does not present any advantage over the API in the dissolution process.

This dissolution rate behavior is similar to that observed in a preliminary study conducted for a Ciprofloxacin–resorcinol cocrystal [30]. In fact, both APIs belong to the same family, and they have similar structures and p*K_a_* values, so similar dissolution rate behavior was expected.

In order to determine whether the dissolution behavior of the cocrystal is the same in a dynamic process along the GIT pH levels, the dissolution test was repeated while monitoring the dissolution in a full sequence of pH values (starting from 2.0 up to 7.4), remaining for 30 min at each pH value. Figure 6 shows the results obtained.

Here, almost complete dissolution was obtained at pH 2.0, and the dissolved amount was the same at pH 4.0 and 5.5. Notice that, in this test, almost all the API was in the solution already when the pH changed from 2.0 to 4.0, so the pH_x=0_ effect was not present. As the API is completely positively charged at pH 4.0, the concentration does not change. When pH 5.5 is reached, around 12% of the compound is already in the zwitterionic form (Appendix A), but the concentration of Norfloxacin in the solution is maintained. When the pH increases to 7.4, there is a sudden precipitation caused by the decrease in solubility at this pH, although 80% of the API remains in the solution. It must be pointed out that this kind of test, where the compound is subjected to different pH values over time, should be more similar to the real processes in the GIT.

XRPD analysis showed again the transformation of both solid forms during the dissolution tests, with the same results already observed in the solubility determinations.

## 4. Conclusions

The crystal form of Norfloxacin, a poorly soluble BCS class IV fluoroquinolone antibiotic, has been modified by cocrystallization with resorcinol. The Norfloxacin–resorcinol cocrystal has been obtained in two different crystal forms, one anhydrous and one monohydrate, with the same (1:1) API/coformer stoichiometry. The anhydrous form can be obtained by solvent-mediated transformation in toluene. The monohydrate is obtained when the anhydrous form is in contact with an aqueous solution.

Although the solubility and dissolution rate determinations have been conducted using the anhydrous form, the final results belong to the hydrated cocrystal since transformation occurs during the course of the determination.

Solubility determination indicates a 10-fold increase in the solubility of the cocrystal compared to the solubility of Norfloxacin itself. The dissolution rate at biorelevant GIT pH values shows that dissolution is complete for both (API and cocrystal) at pH 2.0, since Norfloxacin is totally protonated at this pH. Although the dissolution rate at pH 4.0 was expected to be similar to the one at pH 2.0, it decreases markedly due to the pH_x=0_ effect. However, the presence of a coformer in the cocrystal reduces, to some extent, the buffering effect of Norfloxacin, so that the cocrystal dissolves faster than the pure API. Finally, both solid forms present the lowest dissolution rate at pH 7.4, the pH at which API has the minimum solubility. When the dissolution rate of the cocrystal was tested in a dynamic manner, in which the pH changed from 2.0 to 7.4, remaining for 30 min at each biorelevant pH, the percentage of Norfloxacin in the solution was practically 90% from pH 2.0 to pH 5.5. Only when the pH is changed to 7.4 is there a sudden drop in the concentration, although 80% of the total Norfloxacin remains dissolved, which could potentially contribute to complete absorption throughout the gastrointestinal tract upon oral administration.

## Figures and Tables

**Figure 1 pharmaceutics-14-00049-f001:**
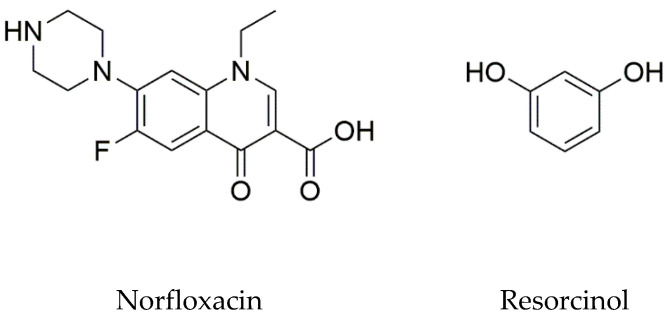
Molecular structures of Norfloxacin and resorcinol.

**Figure 2 pharmaceutics-14-00049-f002:**
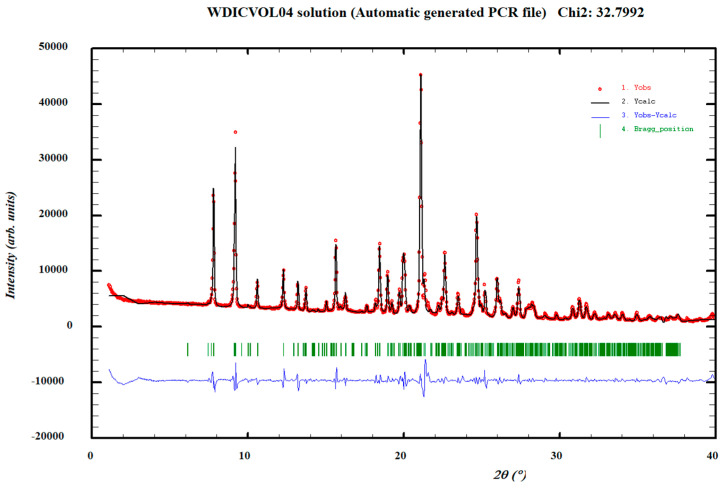
Pattern matching refinement of the Norfloxacin–resorcinol cocrystal. Agreement factors: R_wp_: 9.5%; R_exp_: 1.7% (Chi^2^ = 33). Experimental XRPD profile (red marks), calculated XRPD profile (black solid line) and the difference between them (blue, line). Tick marks correspond to peak positions (|, in green).

**Figure 3 pharmaceutics-14-00049-f003:**
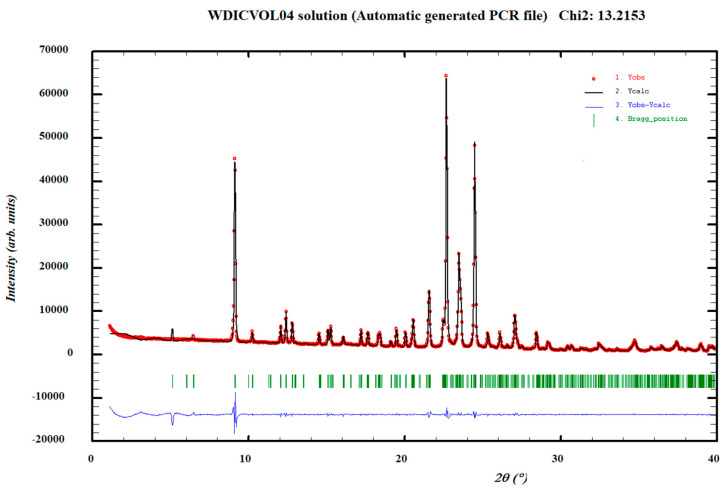
Pattern matching refinement of the Norfloxacin–resorcinol cocrystal monohydrate. Agreement factors: R_wp_: 6.5%; R_exp_: 1.8% (Chi^2^ = 13 Experimental XRPD profile (red marks), calculated XRPD profile (black solid line) and the difference between them (blue, line). Tick marks correspond to peak positions (|, in green).

**Figure 4 pharmaceutics-14-00049-f004:**
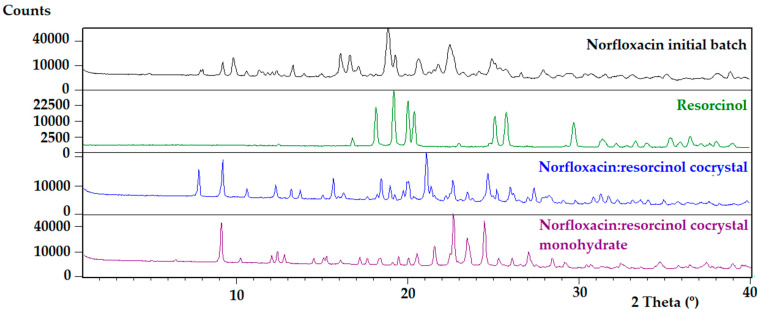
XRPD diffractograms of anhydrous and monohydrate forms of Norfloxacin–resorcinol cocrystal compared to Norfloxacin initial batch and resorcinol.

**Figure 5 pharmaceutics-14-00049-f005:**
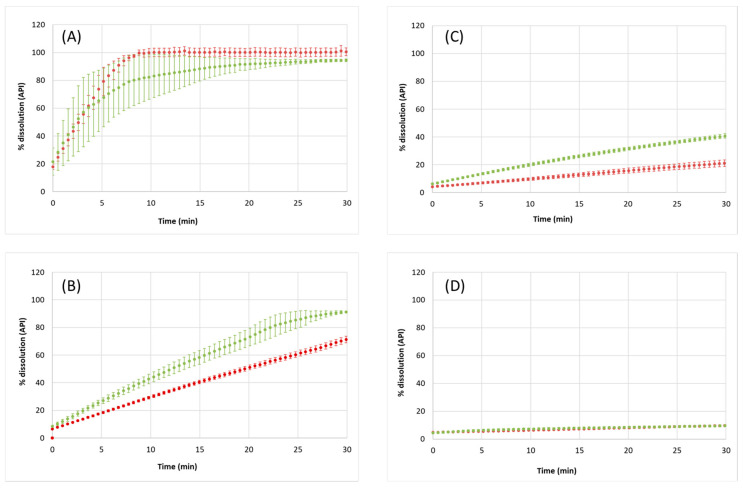
Dissolution profiles of Norfloxacin (•) and Norfloxacin–resorcinol cocrystal (•) at different pH values: pH 2.0 (**A**); pH 4.0 (**B**); pH 5.5 (**C**), and pH 7.4 (**D**).

**Figure 6 pharmaceutics-14-00049-f006:**
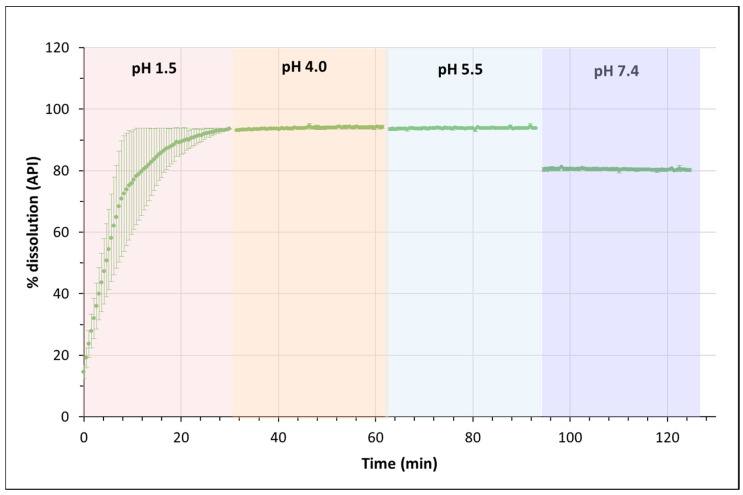
Dissolution profile of the Norfloxacin–resorcinol cocrystal in a 4-sector pH test.

**Table 1 pharmaceutics-14-00049-t001:** Percentage of dissolution and dissolution rate (μg min^−1^) of Norfloxacin and Norfloxacin–resorcinol cocrystal at different GIT pH values. Standard deviations are given in brackets.

Compound	Starting Solid Form	pH	% Dissolved at t = 30 min	Dissolution Rate	Final Solid Form
Norfloxacin	Mixture of polymorphs	2.0	101.2 (6.3)	4.16 (0.73)	Tetrahydrate
		4.0	71.3 (6.9)	0.79 (0.03)	
		5.5	23.0 (5.2)	0.19 (0.02)	
		7.4	9.4 (1.0)	0.06 (0.01)	
Norfloxacin–resorcinol cocrystal	Anhydrous	2.0	94.4 (1.1)	2.53 (1.63)	Monohydrate
		4.0	91.2 (0.9)	0.70 (0.15)	
		5.5	48.0 (12.5)	0.33 (0.04)	
		7.4	9.6 (1.0)	0.19 (0.10)	

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
