# Peer review of "Synthesis and Characterization of a New Norfloxacin/Resorcinol Cocrystal with Enhanced Solubility and Dissolution Profile"

_pharmaceutics, 2021, doi:10.3390/pharmaceutics14010049_

Round 1

Reviewer 1 Report

The manuscript by Fael et al. synthesized a new crystalline form of norfloxacin using resorcinol as conformer at 1:1 molar ratio and characterized the solubility and dissolution rate at different pH conditions. The new crystalline form has shown significantly improved solubility and dissolution compared to the norfloxacin alone. The dissolution study results at different pH conditions and in a media with a gradual change in pH are interesting. In general, the study is well written and easy to follow while the topic covered is interesting to readers. I would recommend accepting this manuscript for publication after a revision for the following comments.

Line 10: Change the font case of ‘Norfloxacin’ throughout the manuscript.

Line 13: Add a space in between the sentences.

Line 37: Correct the spelling of ‘cyclodextrines’.

Line 91: In the abstract, the composition of API and conformer was mentioned as 1:1 which is not matching with the composition mentioned in the section 2.2. Correct.

Section 2.3. and 2.4.: Mention the heating range.

Section 2.7: Solubility studies should be conducted at the human body temperature. Author’s should justify the selection of temperature conditions in their study.

Section 2.7: Add the validation results for the LC method used to quantify API and conformer.

Section 3.1.: Check and update the DSC thermograms and TGA given in the supplementary information as no thermograms were visible.

Section 3.1.: Cite the supplementary information at respective discussion.

 Figure 4: Indicate the characteristic peaks 2 theta position in diffractograms and add y-axis scales.

Line 296: Correct the solubility number given (2.64 not 2,64).

Figure 5: Add error bars to the dissolution profiles and compare them statistically.

Table 1: What are the values given in the brackets? Indicate if they are standard deviation values or standard error means.

Author Response

The manuscript by Fael et al. synthesized a new crystalline form of norfloxacin using resorcinol as conformer at 1:1 molar ratio and characterized the solubility and dissolution rate at different pH conditions. The new crystalline form has shown significantly improved solubility and dissolution compared to the norfloxacin alone. The dissolution study results at different pH conditions and in a media with a gradual change in pH are interesting. In general, the study is well written and easy to follow while the topic covered is interesting to readers. I would recommend accepting this manuscript for publication after a revision for the following comments.

Line 10: Change the font case of ‘Norfloxacin’ throughout the manuscript.

Reply: This has been changed throughout the manuscript.

Line 13: Add a space in between the sentences.

Reply: Space has been added.

Line 37: Correct the spelling of ‘cyclodextrines’.

Reply: Spelling has been corrected

Line 91: In the abstract, the composition of API and conformer was mentioned as 1:1 which is not matching with the composition mentioned in the section 2.2. Correct.

Reply: This does not need to be corrected because it is not a mistake. The composition of the cocrystal is 1:1 indeed, however the synthesis of the cocrystal requires the use of a different stoichiometric ratio (0.30:0.36) in order to be prepared with total purity.

Section 2.3. and 2.4.: Mention the heating range.

Reply: Heating range has been included

Section 2.7: Solubility studies should be conducted at the human body temperature. Author’s should justify the selection of temperature conditions in their study

Reply: It is true that studies at human body temperature would be more realistic. However, the majority of solubility and dissolution rate studies are conducted at 25ºC, so we decided to do it at this reference temperature for further comparison of our values to others already published.

Section 2.7: Add the validation results for the LC method used to quantify API and conformer.

Reply: More details about the quantification parameters have been added at the end of section 2.7.

Section 3.1.: Check and update the DSC thermograms and TGA given in the supplementary information as no thermograms were visible.

Reply: The supplementary information section has been corrected.

Section 3.1.: Cite the supplementary information at respective discussion.

Reply: The supplementary information has been cited as suggested.

Figure 4: Indicate the characteristic peaks 2 theta position in diffractograms and add y-axis scales.

Reply: We have included a new table in the supplementary information section containing the characteristic 2 theta peaks of each new form because we think it provides a clearer information rather than including more markers in figure 4. We hope the referee finds this alternative suitable. On the other hand, the y-axis scales have been included in figure 4.

Line 296: Correct the solubility number given (2.64 not 2,64).

Reply: The solubility number has been corrected.

Figure 5: Add error bars to the dissolution profiles and compare them statistically.

Reply: Error bars were present already in the Figure. However, we have made them more visible than in the original manuscript. Notice that errors at pH 5.5 and 7.5 are very low, and error bars are hardly visible. We have added a comment in Section 3.3.

Table 1: What are the values given in the brackets? Indicate if they are standard deviation values or standard error means.

Reply: Values in brackets are standard deviation values. It is indicated in the table heading.

Reviewer 2 Report

I would like to congratulate Authors for the great work. Co-crystals concept is 2 decade old. Not very widely used unless it brings significant benefit. There are few co-crystals in market. This paper bring dissolution / solubility benefit, but does not bring any benefit. Because Norfloxacin has good solubility already which is 0.32 mg/mL give close to 1 gm max absorption based on solubility. Permeability is challenge and co-crystal  wont help to improve permeability. Plus resorcinol not accepted orally. It helps to prove co-crystal concept. but cannot be used for oral co-crystal. 

Thanks for great work, I would just publish as good co-crystal work without any benefit to bring to practicality. 

Thanks

Sudhakar  

Author Response

Reply: We thank again the reviewer for his careful assessment. We agree with him about the industrial benefit issues. However our manuscript intends to provide with new data to the scientific community rather than offering a practical solution for a particular problem. On the other hand, as stated in the manuscript the European Food Safety Authority, enables resorcinol as an authorized flavoring substance without “safety concern at estimated levels of intake as flavouring substances”.

Reviewer 3 Report

This manuscript reports two new cocrystal compounds of norfloxacin and resorcinol. PXRD, NMR and thermal analyses have been used for confirming the formation of new cocrystal. In general the discussions around the results are well presented with rational behind selection of each method well explained. I did prefer to have single crystal XRD data on the paper to confirm the crystal structure of the new cocrystals. I have some questions below that I would like the authors to address before accepting for publication.

What are the advantages of cocrystallization over the other mentioned techniques for improving dissolution rate?

What is the reason of using slightly higher resorcinol (0.36 mmol) compared to norfloxacin (0.3 mmol)?

DSC curves in SI are broken. Please replace.

It is very interesting to see how the dissolution rate is changing in solutions with various pH values. However, as the dissolution profile of the cocrystal and the API are closely following each other, how do you justify the cocrystallization of this compound in comparison to the pure API in terms of improving the dissolution rate?

Do you have any accelerated stability analysis done on the cocorystal? How does it compare to initial API?

Author Response

This manuscript reports two new cocrystal compounds of norfloxacin and resorcinol. PXRD, NMR and thermal analyses have been used for confirming the formation of new cocrystal. In general the discussions around the results are well presented with rational behind selection of each method well explained. I did prefer to have single crystal XRD data on the paper to confirm the crystal structure of the new cocrystals. I have some questions below that I would like the authors to address before accepting for publication.

Reply: We have tried unsuccessfully our best to get single crystals of enough quality for SCXRD determination. However, in our opinion the crystal cell indexation and refinement in combination with the other techniques confirm the identity of the new cocrystal.

What are the advantages of cocrystallization over the other mentioned techniques for improving dissolution rate?

Reply: Cocrystallization offers several advantages in general, such as being a process with high potential for industrial scale up, it is a method approved by the FDA and EMA as suitable alternatives to existent crystal forms, etc. However, the aim of our work is to provide with new data in the area of cocrystal solubility to the scientific community rather than offering a better solution.

What is the reason of using slightly higher resorcinol (0.36 mmol) compared to norfloxacin (0.3 mmol)?

Reply: We have tried cocrystallization experiments with equimolar amounts of each component but the resulting solids always showed traces of Norfloxacin. However, when using an excess of coformer the resulting solid was pure and the extra resorcinol remained dissolved in the filtered solvent.

DSC curves in SI are broken. Please replace.

Reply: The supplementary information section has been corrected.

It is very interesting to see how the dissolution rate is changing in solutions with various pH values. However, as the dissolution profile of the cocrystal and the API are closely following each other, how do you justify the cocrystallization of this compound in comparison to the pure API in terms of improving the dissolution rate?

Reply: It is true that the profiles of both solid forms are quite similar, although a slightly better dissolution is observed for the cocrystal at pH 4 and 5.5. However, from our point of view, the interest of the cocrystallization is not only the improvements in dissolution rate, but the combined results of a much better solubility and an slightly improved dissolution rate.

Do you have any accelerated stability analysis done on the cocorystal? How does it compare to initial API?

Reply: We agree with the reviewer that this information could also be of interest, but the stability of norfloxacin under temperature / humidity conditions was not a topic that we intended to study in this work.

Reviewer 4 Report

This manuscript synthesized a new cocrystal and investigated the enhancement of solubility and dissolution profile. This topic is crucial in drug design and development. This manuscript fit the scope of Pharmaceutics. I recommend publishing this manuscript after minor revision. My suggestions are listed as below.

  1. Please added a few sentences to describe how to decide toluene as the appropriate solvent. Is that possibility to use other solvent such as ethanol with less toxicity compared with toluene?
  2. The ratio of API and conformer is determined as 1:1 but in the solvent mediated transformation procedure, the molar ratio of API and conformer is 0.30: 0.36. Please also give comment on how to decide this ratio and how to separate the remained excess coformer.
  3. Since some analytical results of cocrystals were presented in supporting information, I still recommend moving PXRD data of pure API and pure conformer to Figure 4 to clearly compare with the PXRD patterns of cocrystals. In addition, please also add the comparison of DSC results of API, cofomer and two cocrystals in the main text. In some case, two solid may produce a eutectic system and show different DSC results compared with two pure solid. Please also give a discussion to exclude this concern.
  4. In Fig. 5, I recommend adding the correlated curve from Eq. (1) and discuss the dissolution rate quantitative using the model parameters such as Kd.

Author Response

This manuscript synthesized a new cocrystal and investigated the enhancement of solubility and dissolution profile. This topic is crucial in drug design and development. This manuscript fit the scope of Pharmaceutics. I recommend publishing this manuscript after minor revision. My suggestions are listed as below.

Please added a few sentences to describe how to decide toluene as the appropriate solvent. Is that possibility to use other solvent such as ethanol with less toxicity compared with toluene?

Reply: Other solvents such as tetrahydrofuran also produced the cocrystal but not totally pure. We eventually decided to use toluene because resorcinol is very soluble in most organic solvents except in toluene, in which its solubility is moderate. Thus, this was our choose since solvent mediated transformation experiments require solvents in which the difference of solubility between both components is as small as possible.

The ratio of API and conformer is determined as 1:1 but in the solvent mediated transformation procedure, the molar ratio of API and conformer is 0.30: 0.36. Please also give comment on how to decide this ratio and how to separate the remained excess coformer.

Reply: We have tried cocrystallization experiments with equimolar amounts of each component but the resulting solids always showed traces of Norfloxacin. However, when using an excess of coformer the resulting solid was pure and the extra resorcinol remained dissolved in the filtered solvent.

Since some analytical results of cocrystals were presented in supporting information, I still recommend moving PXRD data of pure API and pure conformer to Figure 4 to clearly compare with the PXRD patterns of cocrystals.

Reply: We have included PXRD diagrams of pure API and coformer in figure 4 as suggested by the reviewer.

In addition, please also add the comparison of DSC results of API, cofomer and two cocrystals in the main text. In some case, two solid may produce a eutectic system and show different DSC results compared with two pure solid. Please also give a discussion to exclude this concern.

Reply: In our opinion, the cell indexation and further refinement of the new forms are more definitive than any assessment based on DSC. As stated by the reviewer the formation of a eutectic system could produce different DSC traces but the PXRD data discard this possibility. Thus, we kindly ask the reviewer to keep this section as it is.

In Fig. 5, I recommend adding the correlated curve from Eq. (1) and discuss the dissolution rate quantitative using the model parameters such as Kd.

Reply: Notice that curves in Figure 5 are the average experimental curves of a minimum of 3 replicates for each solid form at each pH. The fits of equation 1 to experimental data must be done for each individual curve, and not as an average. That is the reason why the fit is not represented in Figure 5. However, we provide in Table 1 the dissolution rate obtained through the fit. We have not included the rate constant for dissolution, kd, and the extrapolated solubility, S, because in our opinion, just for comparative purposes between the two solid forms, the average experimental curves with the error bars provide the clearest way to realise differences in dissolution behaviour. We would prefer leaving the discussion as it is, if the reviewer agrees with that.  

Reviewer 5 Report

The idea, concept, experimental design, methods and results of the manuscript by Fael et al. seem to be very similar to those of the article "Dissolution rate of ciprofloxacin and its cocrystal with resorcinol” published in ADMET and DMPK in 2018 (https://doi.org/10.5599/admet.6.1.497). Some texts in the manuscript are exactly the same to those in the published article. 

The authors should explain and clarify the similarities and differences between the two documents to avoid duplicate publication and self-plagiarism. 

Author Response

The idea, concept, experimental design, methods and results of the manuscript by Fael et al. seem to be very similar to those of the article "Dissolution rate of ciprofloxacin and its cocrystal with resorcinol” published in ADMET and DMPK in 2018 (https://doi.org/10.5599/admet.6.1.497). Some texts in the manuscript are exactly the same to those in the published article.

The authors should explain and clarify the similarities and differences between the two documents to avoid duplicate publication and self-plagiarism.

Reply: It is true that part of the experimental design of the manuscript related to ciprofloxacin cocrystal is similar to the one of norfloxacin. In fact, preliminar dissolution rate studies were done for both APIs in parallel at the same time in our lab. This is the reason why the experimental procedure related to the dissolution rate determinations is similar in both articles. However, whereas in the case of ciprofloxacin only a preliminar dissolution rate study was done, a much deeper study has been done for norfloxacin. The current manuscript presents a complete characterization from several points of view. Here we show the full solid state characterization for the API and two cocrystal forms, and also provide a complete solubility study which demonstrates that the solubility of the cocrystal is ten-fold higher than the solubility of the API. None of this was done in the case of ciprofloxacin. Moreover, although the dissolution rate results are similar (which is not surprising since both APIs belong to the same family and have similar pKa values), in the present work we provide an explanation based on the pHx=0 concept to understand the better dissolution of the cocrystal (compared to the API) at certain pH values, and all this is supported by additional experiments. All in all, we agree with the referee that a small part which describes the dissolution rate experimental procedures is similar, but in any case duplicate publication or self-plagiarism cannot be argued.

Round 2

Reviewer 5 Report

The authors have clarified and explained the differences between the manuscript and the article "Dissolution rate of ciprofloxacin and its cocrystal with resorcinol” [ADMET & DMPK 6(1) (2018) 61-70; https://doi.org/10.5599/admet.6.1.497], eliminating the possibility of duplicate publication. However, the published article should be mentioned and discussed in more details in the manuscript, such as in the Introduction, or Results and Discussion section, to show new contributions of the manuscript and therefore to justify its publication. 

In addition, some texts in the manuscript are exactly the same to those in the published article, such as in L32-35: “Currently about 60% of the drugs coming from synthesis are poorly soluble and hence have a low oral bioavailability. Diverse strategies have been developed to improve solubility, dissolution rate and subsequently the bioavailability of these drugs.” The authors should check carefully throughout the manuscript and rewrite those texts to avoid self-plagiarism.

Other comments and suggestions to improve the quality of the manuscript are listed below.

1. L120-139 of section 2.6 should be moved to the Results and Discussion section.

2. L382-385: The experimental design to vary the medium pH from 2.0 to 7.4 must be described in the Materials and Methods section.

3. Figure 6: Why the dissolution profile of norfloxacin was not provided to compare with that of the cocrystal? Error bars should be added.

4. L248-250: The procedure to determine the pKa values of norfloxacin (even unpublished) should be described (at least briefly) in the manuscript or in the supplementary information.

5. The authors claimed that “Norfloxacin was a mixture of polymorph A and C (see Figure S1 for further details)” (L74-75). However, some Bragg peaks appeared in the XRD pattern of Norfloxacin initial batch, such as at ca. 10.5 and 13.5o, were not found in the diffractograms of Noflixacin polymorph A and C. Please clarify.

6. L338-341: The authors should discuss why high variability was observed for the dissolution of the cocrystal but not for the API.

7. Table 1 was provided but was not mentioned anywhere in the text.

Author Response

The authors have clarified and explained the differences between the manuscript and the article "Dissolution rate of ciprofloxacin and its cocrystal with resorcinol” [ADMET & DMPK 6(1) (2018) 61-70; https://doi.org/10.5599/admet.6.1.497], eliminating the possibility of duplicate publication. However, the published article should be mentioned and discussed in more details in the manuscript, such as in the Introduction, or Results and Discussion section, to show new contributions of the manuscript and therefore to justify its publication. 

Reply: the manuscript "Dissolution rate of ciprofloxacin and its cocrystal with resorcinol” [ADMET & DMPK 6(1) (2018) 61-70 has been mentioned in the introduction and also is cited and compared to the present work a couple of times in the results and discussion section.

In addition, some texts in the manuscript are exactly the same to those in the published article, such as in L32-35: “Currently about 60% of the drugs coming from synthesis are poorly soluble and hence have a low oral bioavailability. Diverse strategies have been developed to improve solubility, dissolution rate and subsequently the bioavailability of these drugs.” The authors should check carefully throughout the manuscript and rewrite those texts to avoid self-plagiarism.

Reply: the text indicated by the reviewer has been slightly modified, as suggested. Other parts in materials and methods section have been also slightly modified.

Other comments and suggestions to improve the quality of the manuscript are listed below:

  1. L120-139 of section 2.6 should be moved to the Results and Discussion section.

Reply: L120-139 of section 2.6 has been moved to the Results and Discussion section.

  1. L382-385: The experimental design to vary the medium pH from 2.0 to 7.4 must be described in the Materials and Methods section.

Reply: the procedure has been included in section 2.8.

  1. Figure 6: Why the dissolution profile of norfloxacin was not provided to compare with that of the cocrystal? Error bars should be added.

Reply: Figure 6 only shows the dissolution of the cocrystal because our intention was now to see the different dissolution behaviour of the cocrystal when the sequential pH experiments (which model better the dynamic process along the GIT) are performed, compared to dissolution at individual pH values. Comparison with norfloxacin was already done in Figure 5. However, we have now added the error bars in Figure 6.

When norfloxacin is analysed in a 4-sector experiment behaves in the same way as the cocrystal: the compound is completely solved at pH 2 and stays in solution until concentration suddenly drops at pH 7.4 due to the minimum solubility at this pH. In our opinion adding norfloxacin profile in Figure 6 does not contribute in a significant way to the discussion so we kindly ask the reviewer to leave it only with the cocrystal.  

  1. L248-250: The procedure to determine the pKa values of norfloxacin (even unpublished) should be described (at least briefly) in the manuscript or in the supplementary information.

Reply: as suggested, a new section (2.9) has been created to briefly explain the pKa determination procedure.

  1. The authors claimed that “Norfloxacin was a mixture of polymorph A and C (see Figure S1 for further details)” (L74-75). However, some Bragg peaks appeared in the XRD pattern of Norfloxacin initial batch, such as at ca. 10.5 and 13.5o, were not found in the diffractograms of Noflixacin polymorph A and C. Please clarify.

Reply: We agree with the reviewer. The starting material also contained the sesquihydrate of norfloxacin (refcode PUZGAT). This has been corrected in section 2.1 and in figure S1 accordingly.

  1. L338-341: The authors should discuss why high variability was observed for the dissolution of the cocrystal but not for the API.

Reply: we attribute the lower reproducibility in the dissolution of the cocrystal at pH 2 to the different physical properties of the two tablets. Notice that at this pH dissolution is very fast, and Norloxacin and cocrystal powders are physically different. When dissolution is very fast, tablets may dissolve in a heterogeneous way, and it seems that in some way the dissolution of the cocrystal is more affected than the dissolution of the API. We have added a small discussion in the text.

  1. Table 1 was provided but was not mentioned anywhere in the text.

Reply: Table 1 is now cited in the first paragraph (5th and 7th line) of section 3.3, dissolution rate study.
